# Porin A and α/β-hydrolase are necessary and sufficient for hemolysis induced by *Bartonella bacilliformis*

Alexander A. Dichter[1], Florian Winklmeier[1], Diana Munteh[1], Wibke Ballhorn[1], Sabrina A. Becker[1], Beate Averhoff[2], Halvard Bonig [3], Adrian Goldman [4], Meritxell García-Quintanilla [1], Luis Solis Cayo [5], Pablo Tsukayama [5] & Volkhard A. J. Kempf [1]✉

Carrion's disease is endemic to the South American Andes and is characterized by fatal hemolytic anemia. This neglected tropical disease is caused by *Bartonella bacilliformis*, a fastidious and slow-growing pathogen difficult in genetic manipulation. In this study, we determine that porin A and α/β-hydrolase are both necessary and sufficient for hemolysis induced by *B. bacilliformis*. These genes are identified through a screen of a Tn5 transposon mutant library. Using markerless deletion mutagenesis, porin A and α/β-hydrolase deletion mutants are generated and functionally analyzed by hemolysis assays. In silico analyses predict conserved biological functions and three-dimensional structures of the identified proteins, with the α/β-hydrolase showing structural similarity to known lipases. Site-directed mutagenesis of the α/β-hydrolase active site demonstrates that the catalytic triad (Ser205, Asp267, His310) is essential for its hemolytic function. Screening of a phospholipase inhibitor library comprising 27 bioactive compounds identifies compound 48/80 as a potent inhibitor of hemolysis, with activity in the micromolar range. Unraveling the molecular mechanisms underlying Carrion's disease may facilitate the future development of anti-virulence therapies, a promising strategy particularly in the context of increasing antibiotic resistance of *B. bacilliformis*.

Carrion's disease is a neglected vector-borne tropical disease endemic to the Andean valleys of South America, particularly in Peru. It is caused by *Bartonella bacilliformis*, a strictly human-specific, Gram-negative, facultative intracellular bacterium transmitted by *Lutzomyia* sand flies. The disease progresses in two phases: the acute "Oroya fever," marked by massive erythrocyte invasion and severe hemolytic anemia with untreated lethality rates of up to 88%,

and the chronic phase, characterized by angioproliferative skin lesions ("*verruga peruana*") resulting from infection of endothelial cells[1].

Currently, ciprofloxacin is the recommended treatment for Oroya fever[2], however, emerging antimicrobial resistance is a growing concern. Notably, 26% of isolates from previous outbreaks exhibited ciprofloxacin resistance[3].

[1]Institute for Medical Microbiology and Infection Control, University Hospital, Goethe University, Frankfurt am Main, Germany. [2]Molecular Microbiology & Bioenergetics, Institute of Molecular Biosciences, Goethe University, Frankfurt am Main, Germany. [3]Institute for Transfusion Medicine and Immunohematology, Faculty of Medicine, Goethe University, Frankfurt am Main, Germany. [4]Molecular and Integrative Biosciences, Biological and Environmental Sciences, University of Helsinki, Helsinki, Finland. [5]Laboratorio de Genómica Microbiana, Facultad de Ciencias e Ingeniería & Instituto de Medicina Tropical "Alexander von Humboldt", Universidad Peruana Cayetano Heredia, Lima, Peru. ✉e-mail: volkhard.kempf@unimedizin-ffm.de

The pathogenesis of most *Bartonella* species typically involves erythrocyte invasion, intracellular replication, and long-term bacteremia, usually with minimal hemolysis and low infection rates [e.g., *B. quintana*: 0.005%][4]. These features have been extensively studied in animal models employing rodent-adapted strains such as *B. tribocorum* or *B. taylorii*[5,6]. In contrast, *B. bacilliformis* demonstrates exceptionally high erythrocyte infection rates (60–100%) and profound hemolysis[7,8].

The mechanism of erythrocyte invasion by *B. bacilliformis* remains poorly understood. In vitro studies indicate that bacterial penetration of the erythrocyte membrane occurs several hours after the appearance of membrane indentations, suggesting an active, bacterium-driven process resembling forced endocytosis. This process is likely facilitated by invaginations of the erythrocyte membrane carrying attached bacteria into the host cytoplasm. Such a mechanism may result from a synergistic effect involving the outer membrane invasion-associated locus proteins A and B (IalA/IalB) and the flagellum-mediated bacterial motility (as reviewed in ref. 9). While the formation of intra-erythrocytic vacuoles has been observed[10], free intracellular bacteria have also been documented. Both IalA and IalB are proposed to facilitate erythrocyte invasion, although their exact functions remain unclear[11]. Flagella have been implicated in erythrocyte internalization through the exertion of mechanical force[12]. Host membrane proteins, including glycophorin B and band 3, established receptors for *Plasmodium falciparum*[13,14], may serve as entry mediators[15].

Hemolytic pathogens utilize a range of lytic mechanisms, including pore-forming and enzymatic hemolysins. For instance, *Staphylococcus aureus* α-hemolysin assembles into a homo-heptameric pore that disrupts erythrocyte membranes[16]. *Pseudomonas aeruginosa* secretes PlcH, a phospholipase C that enzymatically degrades membrane phospholipids[17] and the phospholipase A of *Neisseria gonorrhoeae* exhibits a contact-dependent lytic activity on host cell membranes[18]. A landmark study demonstrated that *B. bacilliformis* induces contact-dependent hemolysis via a non-secreted, proteinaceous factor that functions independently of host cell proteins or metabolic activity. However, the specific hemolysin responsible remained yet to be identified[19].

Despite its clinical relevance, Carrion's disease remains poorly understood. Erythrocyte invasion and hemolysis are central to its pathogenesis and largely account for the high lethality associated with the acute phase. A deeper understanding of the underlying virulence mechanisms is therefore critical for the development of targeted therapeutic approaches, particularly in the context of increasing antibiotic resistance. In this study, we report the identification of the hemolytic factors of *B. bacilliformis*, making them potential targets for anti-virulence therapies.

## Results

### Development of a hemolysis assay for *Bartonella bacilliformis* infections

Hemolytic anemia is the most severe clinical manifestation of Oroya fever and results from the infection of erythrocytes by *B. bacilliformis*. To investigate *B. bacilliformis*-induced hemolysis in more detail, an in vitro hemolysis assay was adapted[19]. Freshly isolated human erythrocytes were infected with *B. bacilliformis* KC583 at several multiplicities of infection (MOIs), and hemolysis was assessed both visually and via spectrophotometric quantification of hemoglobin release. The results demonstrated a clear, dose-dependent hemolytic effect after an infection period of 20 h (Fig. 1). The exact quantification of the released hemoglobin (ranging from 32 or 53 mg/l to 1979 or 2214 mg/l; suppl. Figure 1A) and of the intraerythrocytic lactate dehydrogenase (LDH; ranging from 8 or 11 U/l to 276 or 334 U/l; suppl. Figure 1B) provided independent confirmation of hemolysis. Based on these findings, subsequent experiments were performed using an infection setup involving $10^7$ erythrocytes and a MOI of 5–10.

### Exploratory analysis of the nature of hemolytic properties of *B. bacilliformis*

The hemolytic activity of *B. bacilliformis* was previously reported to be contact-dependent, possibly involving a non-secreted, proteinaceous factor[19]. To determine whether these effects were reproducible with our *B. bacilliformis* strain and in our adapted hemolytic assay, we investigated whether heat-treated bacteria, cell lysates, or direct contact of bacteria with erythrocytes are necessary to induce hemolysis. For this purpose, erythrocytes were exposed to heat-treated or sonicated bacteria for 20 h. Mild heat-treatment (65 °C) of bacteria resulted in reduced hemolytic activity, while harsh heat treatment (95 °C) completely abolished hemolysis. Sonicated bacterial lysates (which retained native protein structures) maintained hemolytic capacity similar to untreated wild-type bacteria (Fig. 2A). To assess whether hemolysis requires direct contact between *B. bacilliformis* and erythrocytes, a 0.4 μm pore-size filter was used to physically separate the erythrocytes from the bacteria. No hemolysis was observed under these conditions, suggesting that direct contact between *B. bacilliformis* and erythrocytes is essential for hemolytic activity, thereby implying the absence of a secreted extracellular hemolytic compound (Fig. 2B). These results suggest that *B. bacilliformis*-induced hemolysis is a contact-dependent process mediated by one or more heat-labile,

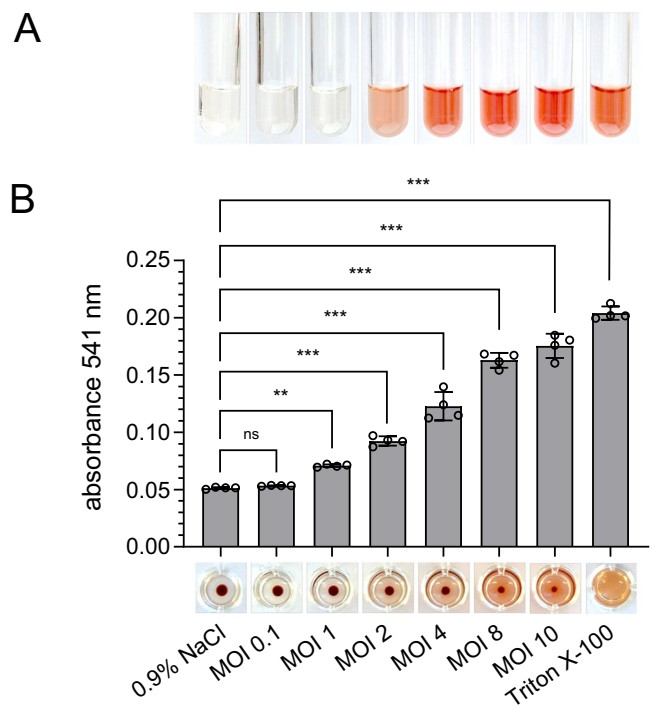

A

B

**Fig. 1 | Development of an in vitro assay to analyze hemolysis of human erythrocytes by *B. bacilliformis*.** Freshly isolated human erythrocytes were washed three times and infected with *B. bacilliformis* strain KC583 (MOIs: 0.1, 1, 2, 4, 8, 10) for 20 h in two independent experiments. Negative control: erythrocytes incubated in 0.9% NaCl without bacteria, positive control: erythrocytes treated with 1% (v/v) Triton X-100. **A** Hemolysis assay supernatant in plastic tubes. Following centrifugation, hemolysis is visible by the amount of hemoglobin released into the supernatant. **B** Hemolysis assay performed in 96-well plate format. Hemoglobin release was quantified photometrically at 541 nm (upper panel) and representative images of erythrocyte pellets in the 96-well plate are shown illustrating the degree of lysis (lower panel). All samples were run in quadruplicates ($n = 4$). Data are presented as mean values ± SD. Statistical significance was determined using one-way ANOVA. Statistical significance was determined by one-way ANOVA (F (7, 24) = 307, $p < 0.001$, $R^2 = 0.989$), followed by Dunnett's post hoc test. ($p \geq 0.01$: not significant, ns; $p < 0.01$: significant **; $p < 0.001$: significant ***). Three independent experiments have been replicated.

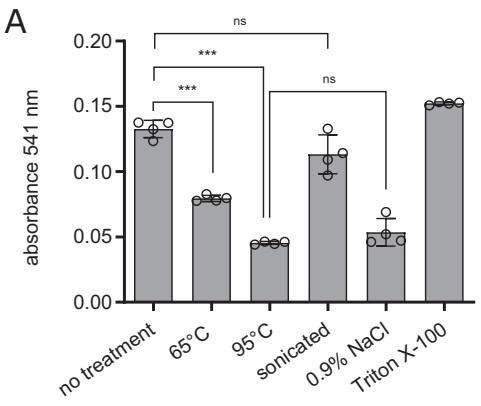

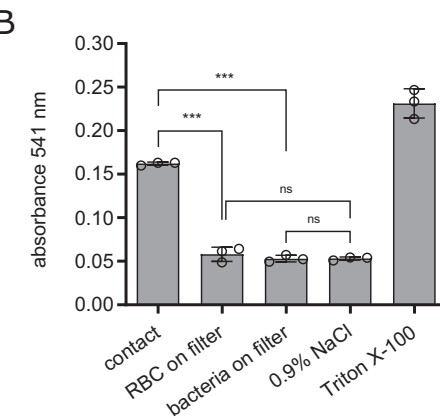

**Fig. 2 | Exploratory analysis of the nature of *B. bacilliformis*-induced hemolysis.** Hemolysis was quantified photometrically at 541 nm after 20 h. **A** Hemolytic activity of bacterial lysates following heat treatment or sonication. Erythrocytes were exposed to bacterial lysates (based on an MOI of 5) or to heat-treated bacteria (65 °C or 95 °C, MOI: 5). **B** Hemolysis assay with *B. bacilliformis* and erythrocytes (based on an MOI of 5) were physically separated by a 0.4 μm pore-size filter in two settings: (i) erythrocytes placed on top of the filter immersed in bacterial suspension (RBC on filter), and (ii) bacteria placed on the filter immersed in erythrocyte suspension (bacteria on filter). Negative control: erythrocytes incubated in 0.9% NaCl without bacteria. Positive control: erythrocytes treated with 1% (v/v) Triton X-100. Data are presented as mean values ± SD. Statistical significance was assessed using one-way ANOVA for (**A**) ($n = 4$, F (5, 18) = 117.6, $p < 0.001$, $R^2 = 0.970$), followed by Tukey's multiple comparison test and for (**B**) ($n = 3$, F (4, 10) = 268.5, $p < 0.001$, $R^2 = 0.991$) followed by Tukey's multiple comparison test ($p \geq 0.01$: not significant, ns; $p < 0.001$: significant ***). Two independent experiments have been replicated.

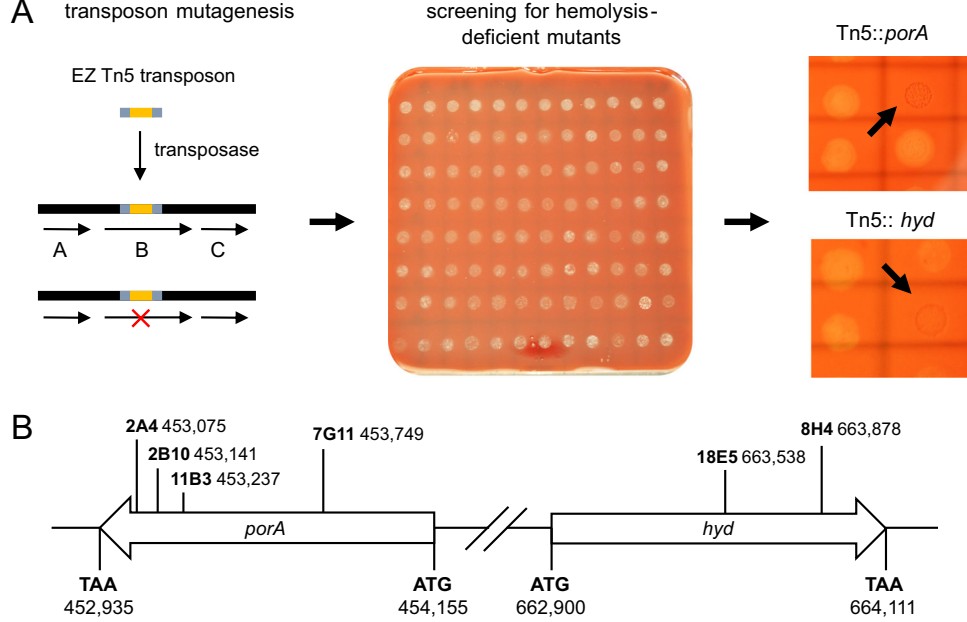

**Fig. 3 | Screening of a *B. bacilliformis* Tn5 transposon library for hemolysis-deficient mutants.** **A** Schematic overview of the Tn5 library construction and hemolysis screening process of a total of 1728 *B. bacilliformis* mutants. Representative CBA hemolysis screening plate displaying 96 transposon mutants after 20 days of incubation. Enlarged view: section of two screening agar plates (illuminated) showing hemolysis-deficient mutants (indicated by arrows). Six mutants were identified, each later confirmed to harbor disruptions in either the *porA* or *hyd* gene locus. **B** Genomic positions of *porA* and *hyd*, with corresponding Tn5 insertion sites.

proteinaceous factors and that this process does not require viable bacteria. Moreover, additional experiments suggest that the hemolytic response mediated by *B. bacilliformis* is independent of the ABO blood group (supplementary Fig. 2).

## Screening of a *B. bacilliformis* transposon library for hemolysis-deficient mutants

To identify potential virulence factors contributing to the hemolytic activity of *B. bacilliformis*, a Tn5 transposon mutant library was generated for strains KC583 and KC584[20], and screened for hemolysis-deficient mutants (Fig. 3A). Following electroporation, mutants were cultivated on Columbia blood agar plates and individual colonies were isolated and transferred into 96-well plates. Transposon mutagenesis yielded a total of 1728 insertion mutants which were screened for loss of hemolytic activity after 14 days of incubation. Mutants lacking hemolysis were subjected to a second screening on human blood agar plates, leading to the identification of six hemolysis-deficient mutants that showed no visible hemolytic activity against human erythrocytes. Sequencing of the transposon insertion sites of these hemolysis-deficient mutants revealed disruptions in two gene loci (reference

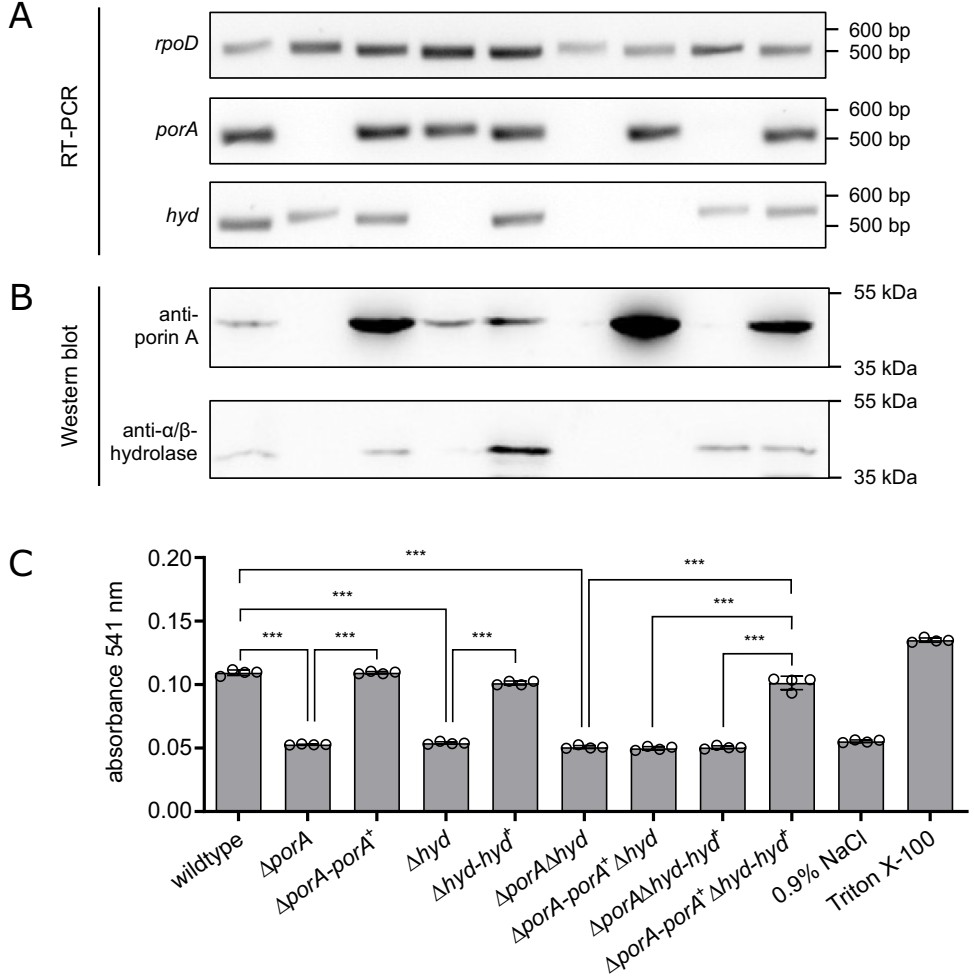

**Fig. 4 | Characterization and functional analysis of *B. bacilliformis porA* and *hyd* deletion mutants.** The following bacterial strains and mutants were used: *B. bacilliformis* wildtype, Δ*porA*, Δ*porA-porA*⁺, Δ*hyd*, Δ*hyd-hyd*⁺; Δ*porA* Δ*hyd*; Δ*porA-porA*⁺ Δ*hyd*, Δ*porA* Δ*hyd-hyd*⁺, Δ*porA-porA*⁺ Δ*hyd-hyd*⁺. **A** Analysis of *porA* and *hyd* gene expression by RT-PCR. For control, *rpod* was used as a housekeeping gene. **B** Analysis of porin A and α/β-hydrolase expression by Western blot analysis using porin A or α/β-hydrolase-specific rabbit antibodies. **C** Hemolytic phenotype of *B. bacilliformis* mutants quantified photometrically at 541 nm after 20 h. Negative control: erythrocytes incubated in 0.9% NaCl without bacteria, positive control: erythrocytes treated with 1% (v/v) Triton X-100. All samples were run in quadruplicate ($n = 4$). Data are presented as mean values ± SD. Statistical significance was determined by one-way ANOVA (F (8, 27) = 687.1, $p < 0.001$, $R^2 = 0.995$), followed by Tukey's multiple comparisons test for correction of multiple comparisons ($p \geq 0.01$: not significant, ns; $p < 0.001$: significant ***). Two independent experiments have been replicated.

sequence NC_008783) in four mutants, the transposon was inserted into BARBAKC583_RS02155 (KC583: 2B10, 7G11, 11B3, KC584: 2A4), and in two mutants into BARBAKC583_RS03125 (KC583: 8H4, 18E5) (Fig. 3B). The BARBAKC583_RS02155 locus encodes a 406 amino acid (aa) porin derived from a 1221-nucleotide open reading frame, hereafter referred to as *porA* or porin A (WP_005766494.1). The BARBAKC583_RS03125 locus encodes a 403 aa α/β-hydrolase from a 1212-nucleotide open reading frame, hereafter referred to as *hyd* or α/β-hydrolase (WP_005766862.1). Both genes possess individual promoters, are not part of operons, and are located in distinct genomic regions, separated by 208,740 nucleotides. Annotation and predicted functions of these loci are based on the *B. bacilliformis* KC583 reference genome (NC_008783.1), as assigned by the NCBI Prokaryotic Genome Annotation Pipeline (see below).

### Generation of markerless *B. bacilliformis porA* and *hyd* deletion mutants and functional analysis

To confirm an essential role of porin A and the α/β-hydrolase in hemolysis and to exclude potential downstream effects resulting from transposon insertion, a markerless gene deletion protocol for *B.*

*bacilliformis* was established (see Materials and Methods) and *porA* and *hyd* were deleted via markerless targeted mutagenesis. The resulting single and double deletion mutants (Δ*porA*, Δ*hyd*, and Δ*porA* Δ*hyd*) were subsequently complemented by transformation with pBBR1MCS-2 plasmids harboring *porA*, *hyd*, or both genes under the control of their respective native promoters (see supplementary Fig. 3). Successful gene deletion and complementation were confirmed by PCR followed by sequencing, RT-PCR, and Western blot analysis using porin A- and α/β-hydrolase-specific rabbit antibodies (see Materials and Methods). The data confirmed successful generation of both single and double deletions as well as corresponding complementation of the targeted genes (see Fig. 4A, B). Notably, α/β-hydrolase was not detected by Western blot in the Δ*porA* mutant, despite expression of *hyd* mRNA (as demonstrated by RT-PCR). This finding is in line with the observation that the α/β-hydrolase is likewise not expressed by *B. bacilliformis* Tn5::*porA* (7G11) (see supplementary Fig. 4).

The hemolytic phenotype of the generated deletion and complementation mutants was assessed by visual inspection of hemolysis on CBA plates and by spectrophotometric quantification of

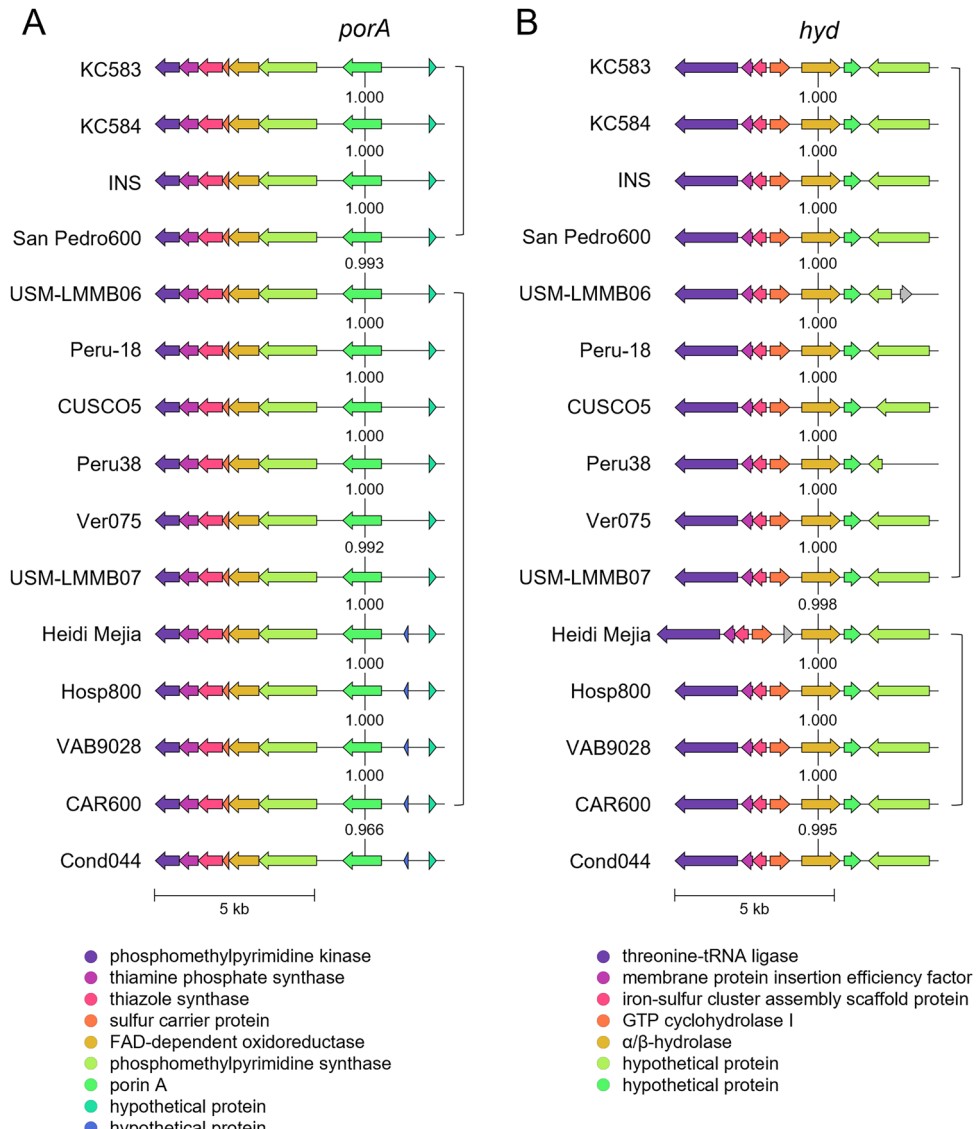

**Fig. 5 | CLINKER-based comparison of gene clusters containing *porA* or *hyd* in *B. bacilliformis*.** For comparative genomic analysis, publicly available complete genome sequences of 15 *B. bacilliformis* isolates were analyzed. Gene clusters were aligned relative to the position of either (**A**) *porA* (green) or (**B**) *hyd* (orange). The predicted amino acid identity of porin A and α/β-hydrolase proteins across isolates is indicated. All isolates display a conserved cluster architecture, with conserved synteny and high sequence similarity highlighting the evolutionary stability of these genomic loci.

hemoglobin release using the hemolysis assay described above. Deletion of either *porA* or *hyd* alone was sufficient to completely abrogate *B. bacilliformis*-mediated hemolysis (Fig. 4C). Complementation with the respective genes (Δ*porA*-*porA*⁺; Δ*hyd*-*hyd*⁺) restored hemolytic activity to levels comparable to the wild-type strain. These results were further corroborated by analyses of the double deletion mutant (Δ*porA* Δ*hyd*) and the corresponding single and double complementation strains (Δ*porA*-*porA*⁺ Δ*hyd*; Δ*porA* Δ*hyd*-*hyd*⁺; Δ*porA*-*porA*⁺ Δ*hyd*-*hyd*⁺) revealing that both proteins in combination are critically required for hemolysis.

**In silico characterization of porin A and the α/β-hydrolase**
Next, we performed in silico analyses on the porin A and α/β-hydrolase. BLASTp revealed that both proteins occur in all analyzed *Bartonella* genomes and that the sequence identity of porin A was approximately 64.5% in the human-pathogenic species *B. henselae* and *B. quintana*, and 60.3% in the rodent pathogen *B. tribocorum* (see supplementary Tables 1, 2). For the α/β-hydrolase, sequence identity

was consistently approximately 79.7% across these species. Within the *B. bacilliformis* species, comparative gene cluster analysis using CLINKER demonstrated that the *porA* and *hyd* loci are conserved across all 15 analyzed *B. bacilliformis* genomes, displaying identical synteny and gene orientation (Fig. 5) indicating strong evolutionary conservation and a likely conserved functional role in hemolysis. When subjected to a multiple sequence alignment using Clustal Omega, the resulting identity matrices demonstrated a high degree of sequence identity in all 15 *B. bacilliformis* genomes for porin A (96.5–100%) and α/β-hydrolase (99.5–100%) (supplementary Tables 3 and 4).

The physicochemical properties of porin A and the α/β-hydrolase were determined using ExPASy ProtParam revealing that both proteins are predicted to be thermally stable (supplementary Table 5). Subcellular localization predictions were generated using PSORTb, CELLO, LocTree3, and SignalP 6.0 (suppl. Table 6) classifying porin A as an outer membrane protein. In contrast, predictions for the α/β-hydrolase were inconsistent: CELLO suggested periplasmic localization, LocTree3 indicated a secreted protein, and PSORTb returned no

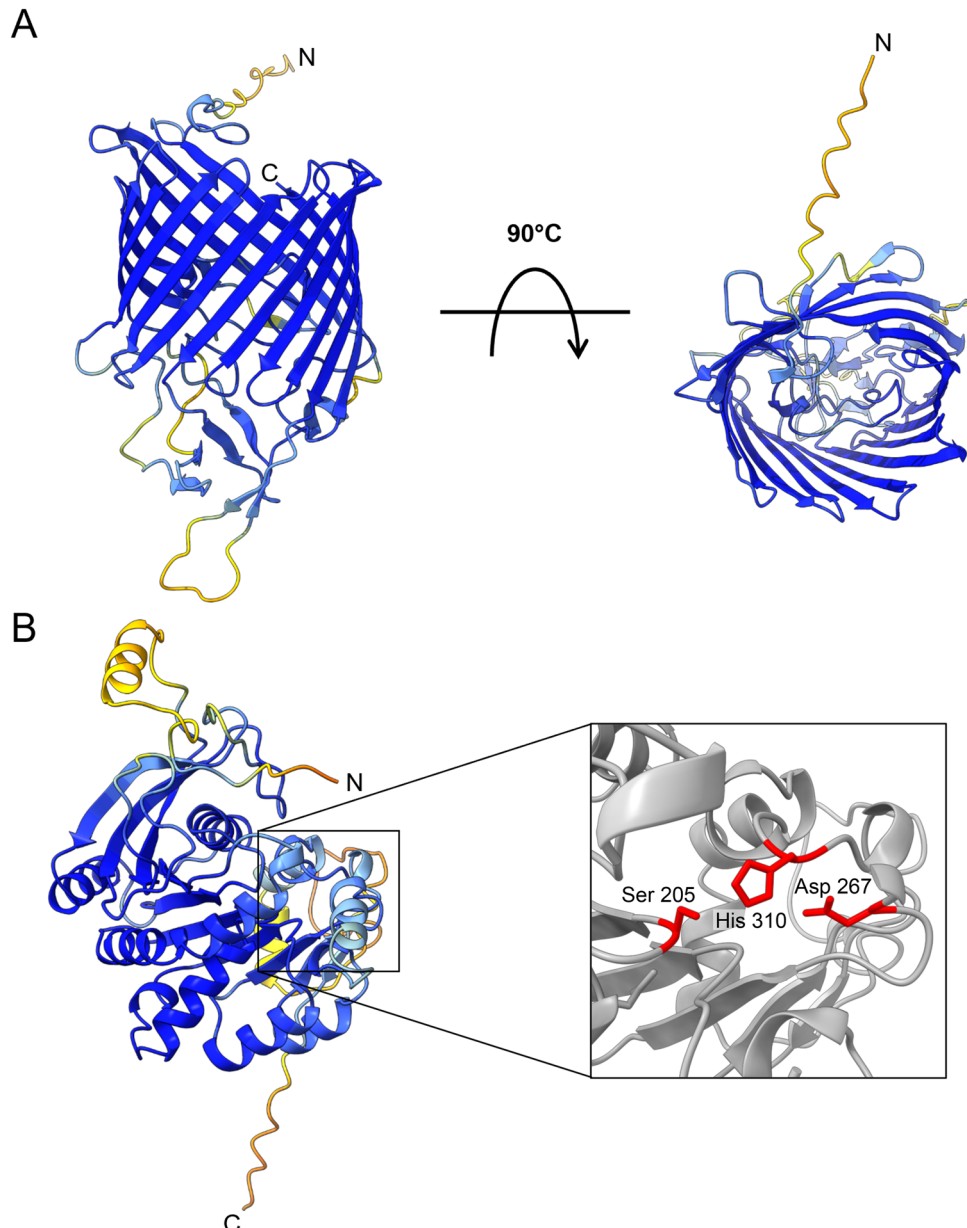

**Fig. 6 | AlphaFold2 structure prediction models of porin A and α/β-hydrolase of *B. bacilliformis*. A** Ribbon representation of porin A, showing a characteristic β-barrel motif predicted with high confidence and **B** of the α/β-hydrolase, displaying a central α/β-sheet fold. The catalytic triad consisting of Ser205, Asp267, and His310 is highlighted. The pLDDT scores of the structure models are color-coded (dark blue: 100–90, light blue to yellow: 90–70, yellow to orange: 70–50, and orange to red: 50–0). Numbering is according to the mature protein after removal of signal peptide.

definitive result. Genetic analysis via SignalP clearly revealed the presence of a lipoprotein signal peptide arguing for an extracellular localization.

Conserved domains and sequence motifs were analyzed using the Conserved Domain Database (CDD), Pfam, and InterProScan to infer potential biological functions (supplementary Table 7). Porin A was classified as an alphaproteobacterial porin with a characteristic porin-like β-barrel structure. The α/β-hydrolase was assigned to the esterase/lipase superfamily, suggesting a putative role in lipid metabolism or membrane remodeling.

### Prediction of the protein structures of porin A and the α/β-hydrolase

To gain structural insights into porin A and the α/β-hydrolase, in silico structure prediction was performed using ColabFold, yielding five structural models for each protein. The top-ranked models, selected based on the highest pLDDT (predicted Local Distance Difference Test) scores, showed high overall confidence, with the majority of residues scoring above 80 indicating reliable predictions for both backbone and side-chain conformations. Regions with lower pLDDT values (<70), such as the N- and C-terminus and residues 59–64, 102–104, 173–182 and 196–201 in porin A (supplementary Fig. 5) or 48–52, 82–89 and 269–281 in the α/β-hydrolase (supplementary Fig. 6) were interpreted with caution due to the decreased model reliability.

The predicted structure of porin A revealed a 16-stranded β-barrel motif forming a transmembrane pore (Fig. 6A), consistent with the above presented analyses of conserved domains and sequence motifs. In the case of the α/β-hydrolase, a characteristic α/β-fold was predicted as the structural core, comprising nine β-strands. With the exception of β1/β2 and β2/β3, each β-strand pair is interconnected by α-helices

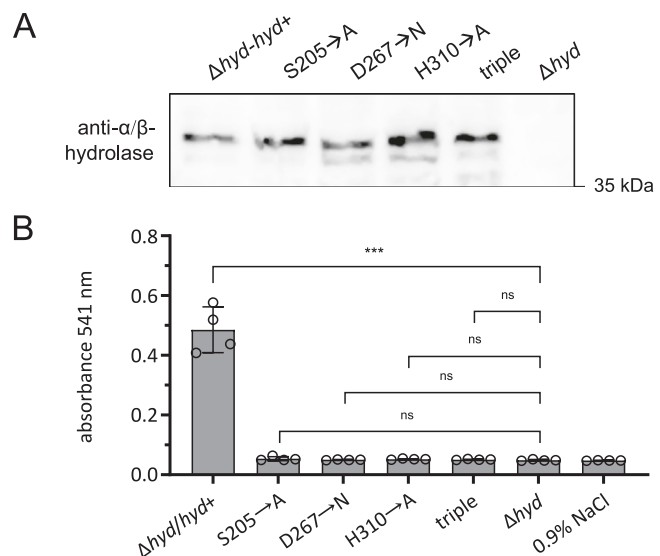

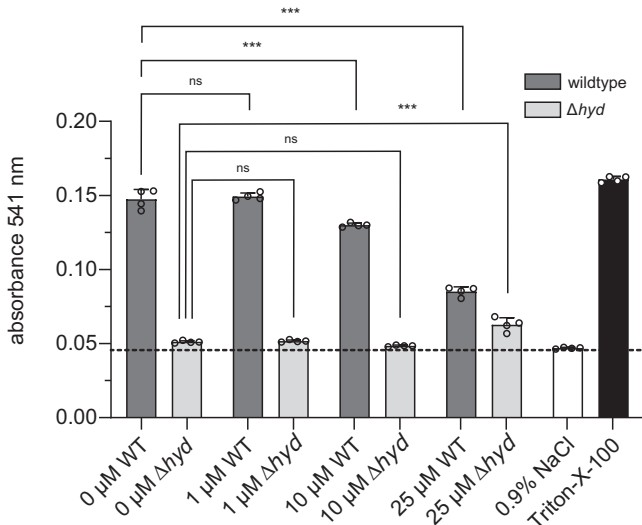

**Fig. 7 | Hemolytic activity of α/β-hydrolase catalytic triad mutants of *B. bacilliformis*. A** Western blot to confirm expression of α/β-hydrolase active site mutations (S205→A, D267→N, H310→A, and the triple mutant) using anti-α/β-hydrolase antibodies. **B** Human erythrocytes were infected (MOI: 5) with *B. bacilliformis* Δ*hyd*/*hyd*⁺ (hemolysis positive), *B. bacilliformis* Δ*hyd* (hemolysis negative) and mutant forms of the α/β-hydrolase (S205→A, D267→N, H310→A, and the triple mutant). Hemolysis was quantified photometrically at 541 nm after 20 h. Negative control: erythrocytes incubated in 0.9% NaCl without bacteria, positive control: *B. bacilliformis* Δ*hyd*/*hyd*⁺. All samples were run in quadruplicate ($n = 4$). Data are presented as mean values ± SD. Statistical significance was determined by one-way ANOVA (F (5, 18) = 126.7, $p < 0.001$, $R^2 = 0.972$), followed by Dunnett's post hoc test ($p \geq 0.01$: not significant, ns; $p < 0.001$: significant ***). Two independent experiments have been replicated.

**Fig. 8 | Inhibition of the hemolytic activity of *B. bacilliformis* by phospholipase inhibitors.** Human erythrocytes were infected (MOI: 5) with *B. bacilliformis* wild-type (dark gray bars) and *B. bacilliformis* Δ*hyd* (light gray bars) in the presence of varying concentrations of compound 48/80. Hemolysis was quantified photometrically at 541 nm after 20 h. Negative control (white bar): erythrocytes incubated in 0.9% NaCl without bacteria, positive control (black bar): erythrocytes treated with 1% (v/v) Triton X-100. Statistical significance was determined using one-way ANOVA (individual *p* values are given in the figure). All samples were run in quadruplicate ($n = 4$). Data are presented as mean values ± SD. Statistical significance was determined by one-way ANOVA (F (7, 24) = 761.5, $p < 0.001$, $R^2 = 0.996$), followed Tukey's multiple comparisons test. Individual p-values are indicated in the figure ($p \geq 0.01$: not significant, ns; $p < 0.001$: significant ***). Two independent experiments have been replicated.

(Fig. 6B). This arrangement supports the formation of a catalytic active site typically composed of a catalytic triad, a highly conserved feature among α/β-hydrolases. The triad consists of a nucleophile, an acidic residue, and a histidine[21,22]. In the *B. bacilliformis* α/β-hydrolase, a serine residue at position 205, located between β-strand β6 and α-helix α4 within the consensus motif [A/G/T]XSXG, serves as the nucleophile. The acidic component of the triad is an aspartate at position 267, positioned between β8 and α6. The conserved histidine is located downstream of the final β-strand (β9), at position 310. Antiparallel strand 10 (359–361) in the AlphaFold model is not seen in solved structures of α/β-hydrolase fold proteins, which either finish at β9, like PDB entry 7xrh (*L. acidophilus* feruloyl esterase) or have a parallel β10 and an antiparallel β11, like PDB entry 5ikx (human acetyltransferase core enzyme). In addition, the final 40 residues (335–374) were deleted without affecting activity (see supplementary Fig. 7). The most similar structures identified through DALI, Cofactor, and PDBeFold include a series of yeast and α/β-hydrolase fold bacterial (phospho-)lipases and esterases such as the yeast *Moesziomyces antarcticus* lipases (1lbt and 4k6g), *L. acidophilus* feruloyl esterase (7xrh) and *Janthinobacterium* sp. strain J3 CarC carbazole hydrolase. A TM-align analysis[23] revealed that they all align structurally 190–217 residues with root mean square deviations (rmsds) in the 3.8–4.2 Å range, tm-scores from 0.425 to 0.468 and structure-based sequence identities of 11.4–13.9% including the canonical Ser-His-Asp triad, with the residues in the same place as in the AlphaFold model.

### Mutagenesis of the catalytic triad of the α/β-hydrolase abrogates *B. bacilliformis*-mediated hemolysis

To further characterize the hydrolytic and thereby hemolytic activity of the α/β-hydrolase of *B. bacilliformis*, site-directed mutagenesis was performed targeting the residues of the predicted catalytic triad. Specifically, the following point mutations were introduced: Ser205→Ala, Asp267→Asn, His310→Ala, as well as a triple mutant (Ser205→Ala, Asp267→Asn, His310→Ala). These mutant variants were expressed in the *B. bacilliformis* Δ*hyd* background (see Materials and Methods for details) and production of mutant proteins was verified in a Western blot (Fig. 7A). Human erythrocytes were infected with *B. bacilliformis* Δ*hyd* expressing either wild-type or mutant forms of the α/β-hydrolase (S205A, D267N, H310A, and the triple mutant). After 20 h, hemolysis was quantified spectrophotometrically. *B. bacilliformis* Δ*hyd-hyd*⁺ (hemolysis positive) and *B. bacilliformis* Δ*hyd* (hemolysis negative) served as controls. All mutant strains expressing the substituted versions of the α/β-hydrolase were deficient in hemolysis similar to the Δ*hyd* deletion strain indicating complete loss of function (Fig. 7B). These findings underscore the essential role of the catalytic triad residues in mediating hemolytic activity and further support the critical involvement of the α/β-hydrolase in the hemolytic mechanism of *B. bacilliformis*.

### Screening of a phospholipase inhibitor library for in vitro inhibition of *B. bacilliformis* mediated hemolysis

The observation that deletion of a single gene (*hyd*) and mutagenesis of its catalytic triad abrogate hemolytic activity suggests that the α/β-hydrolase constitutes a potentially druggable phospholipase target for the inhibition of *B. bacilliformis*-induced hemolysis. To identify potential inhibitory compounds, we screened a commercial library consisting of 27 well-characterized phospholipase inhibitors dissolved in water or DMSO (see Materials and Methods and supplementary Table 8 for details). Matrix effects by the solvent were widely excluded by control experiments (data not shown). Erythrocytes were infected with wild-type *B. bacilliformis* in the presence of varying

concentrations of the phospholipase inhibitors. After 20 h, hemolysis was quantified spectrophotometrically. Among the 27 inhibitors tested, compound 48/80 emerged as a potent inhibitor of hemolysis, displaying optimal inhibitory activity at concentrations between 10 and 25 μM. However, this effect was not observed in *B. bacilliformis* Δ*hyd*, although compound 48/80 itself showed a slight hemolytic effect at 25 μM (Fig. 8).

## Discussion

Hemolysis is a life-threatening condition that leads to tissue hypoxia (due to impaired oxygen transport by erythrocytes), renal failure (due to the release of hemoglobin) and disseminated activation of coagulation, ultimately culminating in the death of the patient. Hemolytic Oroya-fever is one of the deadliest infections for humans with case-fatality rates as high as 88%[24], but knowledge about this disease remains scarce. Only 310 entries using the search term "*B. bacilliformis*" are referenced in PubMed (last access, October 2025). The reasons for this unsatisfying body of evidence is (i) the neglect of this disease outside of the endemic regions (Peru) and (ii) the challenging experimental conditions due to the slow growth of this fastidious pathogen, restrictions in cell culture infection models and the unavailability of animal models rendering this infection a true "neglected tropical disease". This is particularly alarming because of the growing antibiotic resistance against the first-line antibiotic ciprofloxacin and because of climate change, as the vector *Lutzomyia* and the unknown ecological niche of the pathogen could spread to other countries and continents[25,26].

Intravascular hemolysis results in the release of hemoglobin into the bloodstream, acute anemia, and reduced oxygen-carrying capacity. Even moderate levels of free hemoglobin (approx. 0.2 g/kg body weight or 2–3% of total Hb) are sufficient to cause life-threatening conditions in humans[27,28]. As shown in this study, hemolysis depends almost linearly on the infectious inoculum of *B. bacilliformis* and can become nearly complete at high bacterial loads (Fig. 1), reaching concentrations of free hemoglobin known to be associated with severe, potentially fatal toxicity.

Only a few human pathogens are erythrocytotropic, albeit none even remotely as efficiently as *B. bacilliformis*. Thus, *Plasmodium* spp. causes malaria infections with parasitemia rates typically below 1% and, in severe cases, occasionally reaching 2–3% or higher[29]. In contrast, the mean percentage of infected erythrocytes in Oroya-fever is 61% when patients are admitted for medical assistance[7]. Although *B. bacilliformis* shows a strong tropism for human erythrocytes, specific associations between AB0 blood groups and the severity or progression of Oroya fever are not described. Unlike *P. falciparum*, where studies suggest that individuals with blood group 0 experience less severe clinical outcomes than those with blood group A (implying a potential, though not definitive, survival advantage)[30], our results exhibited no association of increased hemolysis in any particular AB0 group for *B. bacilliformis* (supplementary Fig. 2). These results are supported by other studies reporting no significant differences in allele frequencies of MNS, Diego, and Duffy blood group systems between the chronic and acute phases of Oroya fever[31].

Understanding the virulence mechanisms underlying hemolysis is critical for elucidating the pathogenicity of *B. bacilliformis* and for the development of future anti-virulence therapies or vaccines. A logical first step is to investigate the presence of hemolysin-encoding genes in *B. bacilliformis* genomes. However, BLAST analyses confirmed clearly the absence of any homologues to well-characterized hemolysins such as α-hemolysin (Hly) from *S. aureus*[32] or HlyA from *Escherichia coli*[33] (data not shown). Nevertheless, several *B. bacilliformis* genes are actually annotated as putative secreted hemolysins (hemolysin-A: KZN22078.1, KZM38023.1; hemolysin secretion protein-D: KZN22169.1, KZM38155.1). These annotations are based solely on hypothetical computational predictions and lack any experimental validation.

Using a two-chamber infection model separated by a membrane permeable to proteins and small molecules but not to bacteria, we showed that *B. bacilliformis*-induced hemolysis is a contact-dependent process, likely mediated by proteinaceous compounds located on the bacterial surface (Fig. 2). These findings support earlier observations in which the exact hemolytic mechanism, however, remained unresolved[19]. To identify the genetic basis of the hemolytic phenotype, we employed transposon mutagenesis and screened two strains (KC583 and KC584) for non-hemolytic mutants. All transposon insertions resulting in a non-hemolytic phenotype were found either in *porA* or *hyd*, indicating a functional link between these two genes in hemolysis (Fig. 3). In bacteria, functionally related genes are often organized into operons to facilitate coordinated regulation[34]. The fact that *porA* and *hyd* are single genes with different promotors located distantly on the chromosome suggests additional, distinct roles in the infection process, though this remains speculative. Nonetheless, both genes are conserved across all analyzed *B. bacilliformis* isolates within highly conserved genomic clusters (Fig. 5). Homologies in other *Bartonella* ssp. show only moderate sequence similarity (supplementary Tables 1 and 2) consistent with the observation that other *Bartonella* spp., such as the human pathogens *B. quintana* or *B. henselae*[35] or the rat pathogen *B. tribocorum*[36] do not induce a pronounced hemolytic anemia in their hosts.

The essential roles of porin A and the α/β-hydrolase in the process of hemolysis was proven by implementing markerless deletion and complemented mutants in hemolysis assays. For this, we established a markerless deletion approach based on the homologous recombination strategy previously used for *Acinetobacter baumannii*[37], conclusively showing that *porA* and *hyd* are both necessary and sufficient to induce hemolysis in human erythrocytes (Fig. 4). The reason why the α/β-hydrolase protein is undetectable in *B. bacilliformis* Tn5::*porA* and the Δ*porA* mutant remains unclear. Complementation of *porA* in the Δ*porA* mutant restored α/β-hydrolase protein expression, as confirmed by Western blot analysis, despite unchanged hydrolase mRNA levels. These findings suggest that the presence of the porin A protein is essential for hydrolase activity. Right now, it can only be speculated whether porin A may be required for the assembly of the hydrolase into a functional hemolytic complex or for its proper localization within the bacterium.

Porins are transmembrane proteins in the outer membrane of Gram-negative bacteria and consist of 8–24 antiparallel aligned β-sheets, which together form a β-barrel motif. The resulting pore forms a water-filled channel that allows diffusion of small hydrophilic compounds through the outer membrane[38]. According to in silico characterization, porin A of *B. bacilliformis* has a typical domain architecture with 16 β-sheets (Fig. 6A). Two purified porins from *Salmonella typhi* have already been shown to be hemolytic. These porins (designated as OmpC and OmpF) have a high structural similarity (TM score 0.677 or 0.687, respectively, analyzed via TM-Align) and integrate into and permeabilize the erythrocyte membrane ultimately leading to erythrocyte lysis[39]. Interestingly, the α-hemolysin of human pathogenic *S. aureus* forms a homo-heptameric transmembrane pore perforating the cell membrane of erythrocytes and other cells permeabilizing the membrane for ions, water, and small molecules, thus leading to cell lysis[40]. However, our data clearly demonstrate that *B. bacilliformis* porin A alone is not sufficient for inducing hemolysis but relies on the interplay with the α/β-hydrolase (see Fig. 4).

The α/β-hydrolases form one of the largest families of structurally related enzymes and exhibit a broad spectrum of catalytic reactions and phylogenetic diversity[41]. All family members have a characteristic structural motif: an α/β-sheet consisting of several β-sheets connected by α-helices and an active center with a diverse catalytic triad similar to the serine proteases, though the structures are completely unrelated[21,22]. The α/β-hydrolase from *B. bacilliformis* also exhibits such a characteristic α/β-sheet motif with a Ser205, Asp267, His310 catalytic triad in the

active site (Fig. 6B). As expected, there is a consensus sequence AXSXG around the Ser nucleophile. Consistent with this, conservative substitutions of the residues in the catalytic triad resulted in a complete loss of hemolytic activity (Fig. 7). Structural analysis of the α/β-hydrolase of *B. bacilliformis* showed that it was particularly homologous to (phospho-)lipases from the α/β-hydrolase family (see above). This suggests that the α/β-hydrolase acts in fact as a phospholipase that destroys erythrocyte membranes, culminating in fatal hemolysis. The hemolytic activity of *B. bacilliformis* is inhibited by the phospholipase inhibitor compound 48/80 (Fig. 8), further indicating that *B. bacilliformis* α/β-hydrolase is a phospholipase and that this activity is required for hemolysis. This fact may be of interest for the prevention of fatal hemolysis in Oroya fever patients, as compound 48/80 is known to inhibit both phospholipase-A2 and -C[42] and might therefore be considered as a lead compound for the development of a hemolysis inhibitor to mitigate the course of the lethal Oroya fever.

Interestingly, porin A was earlier identified as an immunodominant vaccine target[43] making it interesting as a potential vaccine to prevent hemolysis. Both observations might be of high importance because of the extraordinary lethality of Oroya fever and the strongly increasing ciprofloxacin resistance of the pathogen[3], underscoring the urgent need for alternative therapeutic strategies.

Direct pathogen-erythrocyte contact has been considered essential for hemolysis, as it only occurred following centrifugation of pathogens onto erythrocytes[19]. The inhibition of hemolysis by physically separating *B. bacilliformis* from erythrocytes using filter inserts (which is a more rigorous experimental setup) indicates that bacterial adhesion to erythrocyte surfaces is critical for *B. bacilliformis*-induced hemolysis. Flagella and *B. bacilliformis* adhesins (e.g., Bbad) have been proposed to mediate this adhesion[1]. Bbad, a member of the trimeric autotransporter adhesin family, may promote tight bacterial attachment to and invasion into erythrocytes, facilitating subsequent membrane perforation by porin A and α/β-hydrolase. Clearly, a comprehensive characterization of erythrocyte interaction partners of *B. bacilliformis* is required to elucidate the mechanisms underlying bacterial adhesion and hemolysis.

Electron micrographs demonstrate that *B. bacilliformis* is present either in membrane-enclosed vacuoles or free in the cytosol following invasion of erythrocytes[10]. It has been proposed that the hemolytic mechanism of *B. bacilliformis* may facilitate escape from erythrocytes rather than promote their invasion. Notably, *B. bacilliformis* exhibits pronounced hemolytic activity in vitro, whereas, in vivo, it can persist within infected erythrocytes. This discrepancy suggests that modulation of porin A and α/β-hydrolase expression may play a role in regulating the transition from an intraerythrocytic lifestyle to hemolysis.

Taken together with existing literature, our observations suggest the following hypothetical infection strategy: *B. bacilliformis* adheres to human erythrocytes via flagella and possibly adhesins (Bbad, not yet confirmed)[44,45], which ensure bacterial contact to a yet unknown erythrocytic receptor. Erythrocyte invasion is mediated by the IalA/B system[11]. Hemolysis (mediated by porin A and α/β-hydrolase) may occur following the intracellular proliferation of the pathogen, enabling its escape from erythrocytes. This process might not occur immediately but during the later course of infection, suggesting that porin A and α/β-hydrolase activity may be regulated by yet unidentified factors (possibly iron or heme). This facilitates further erythrocyte infection (Oroya fever) and dissemination to secondary niches such as endothelial cells (*verruga peruana*).

To study these processes in vivo, rhesus macaques (*Macaca mulatta*) remain the only validated animal model for Carrion's disease. In a study from 1926, Noguchi demonstrated that rhesus monkeys developed both Oroya fever and *verruga peruana* following *B. bacilliformis* infection[46]. However, due to ethical and logistical constraints, it is not realistic to pursue this model today. By this, the elucidation of *B. bacilliformis* virulence remains restricted to in vitro infection models.

Further efforts are urgently needed to better understand this neglected tropical disease and to mitigate the impact of Oroya fever on Andean populations.

## Methods

### Ethics
Experiments using human erythrocytes were approved by the Institutional Ethics Committee at University Hospital Frankfurt am Main (January 23rd, 2019). Erythrocyte donors provided written informed consent before the study.

### Culture conditions
*B. bacilliformis* was cultivated on Columbia Blood Agar (CBA) plates containing 5% (v/v) sheep blood (Becton Dickinson, Heidelberg, Germany) at 28 °C in a humidified incubator for 10–14 days. Kanamycin-resistant *B. bacilliformis* strains were selected on CBA plates containing 25 μg/mL kanamycin (MP Biomedical Santa Ana, CA). For the analysis of *B. bacilliformis* hemolysis of human erythrocytes, bacteria were cultured on human blood CBA-plates. For this, human whole blood was collected into EDTA monovettes (Sarstedt, Nümbrecht, Germany) and centrifuged at 500 g for 10 min to separate hematocrit and plasma. The supernatant was discarded and the erythrocytes were washed three times in 0.9% NaCl before adding 5% (v/v) to the Columbia blood agar base. *E. coli* was cultured in Luria-Bertani (LB) medium, or on solid LB medium supplemented with 15 g/L agar, and incubated overnight at 37 °C. Liquid cultures were grown on a culture shaker at 120 rpm. Kanamycin was added to the culture medium at a final concentration of 50 μg/mL to select kanamycin-resistant *E. coli* strains (for details see supplementary Table 9).

### Transposon mutagenesis
Transposon mutagenesis was performed on *B. bacilliformis* strains KC583 and KC584 to identify hemolysis mediating pathogenicity factors. For this purpose, the EZ-Tn5 <KAN-2> Tnp Transposome kit (Epicentre, Madison, WI, USA) was utilized[20]. The transposon harbors the *Tn903* kanamycin resistance gene (aminoglycoside 3'-phosphotransferase), flanked by two 19-bp mosaic ends acting as recognition sequences for the transposase. The transformation of competent *B. bacilliformis* cells with the transposomes was carried out by electroporation. For this, *B. bacilliformis* was grown on CBA plates, harvested with a cotton swab and resuspended in 10 mL *Bartonella* liquid medium[47]. Bacteria were centrifuged at 5000 g for 10 min at 4 °C, washed three times with ice-cold 10% (v/v) glycerol (Sigma-Aldrich, Deisenhofen, Germany) and resuspended to a concentration of 10^10 cells/mL. Subsequently, 40 μL of the bacterial suspension were mixed with 1 μL of transposome and 1 μL of TypeOne™ Restriction Inhibitor (Epicentre) in a pre-cooled electroporation cuvette (Bio-Rad, Dreieich, Germany). Bacteria were electroporated using a Pulse Controller II (Bio-Rad) at a field strength of 2.5 kV/cm and a constant capacitance of 25 mF at 200 V, immediately transferred to 1 mL recovery medium and incubated for 1 h at 30 °C with shaking. To prepare the recovery medium, 200 μL sheep red blood cells were mixed with 800 μL sterile distilled water and incubated at 65 °C for 20 min. The lysate was centrifuged at 20,817 g for 15 min, and 500 μL of the supernatant was retained. An equal volume of 5% (w/v) bovine serum albumin (BSA, Sigma-Aldrich) and 5 mM L-methionine (Sigma-Aldrich) were added and the mixture was incubated for 2 min at 65 °C. Subsequently, 9 mL heart infusion broth (Becton Dickinson) was added, and the solution was filter-sterilized using a 0.2 μm syringe filter (Sarstedt). A volume of 100 μL of each transformation mixture were plated on CBA plates with kanamycin and incubated at 28 °C for 10 days. For the preparation of the *B. bacilliformis* transposon library, the transposon mutants were isolated and cryopreserved in 96-deep-well plates using tryptic soy medium (Becton Dickinson) containing 10% (v/v) glycerol (1 mL bacterial suspension per well) at −80 °C (see supplementary Table 9).

## Screening for non-hemolytic mutants

Hemolysis-deficient *B. bacilliformis* mutants were identified using CBA plates supplemented with 5% (v/v) sheep blood and 25 μg/mL kanamycin. Transposon mutants were thawed on ice and, subsequently, 2 μL of each bacterial suspension was spotted onto the prepared CBA plates and incubated at 28 °C for at least 10 days. Mutants exhibiting loss of hemolytic activity were re-tested in a second screening round under identical conditions to confirm the non-hemolytic phenotype. In a third round, the hemolytic activity of the mutants was further confirmed on CBA plates containing 5% (v/v) human blood. The position of the transposon insertion sites in the genomes of hemolysis-deficient mutants was determined by single-primer PCR[48] (details see supplementary Table 10) followed by Sanger sequencing (Microsynth AG, Göttingen, Germany). To identify the genes associated with the hemolytic phenotype, PCR products were sequenced using nested transposon-specific oligonucleotides and mapped to the genome via alignment in Geneious Prime (Dotmatics, Boston, MA, USA). For subsequent deletion and complementation experiments, the operon structure of the identified genes was analyzed using Operon Mapper[49]. In addition, promoter sequences were predicted in silico using BPROM[50] with default settings.

## Genetic manipulation of *B. bacilliformis*

Deletion of porin A (*porA*) and α/β-hydrolase (*hyd*) was carried out by markerless mutagenesis following the protocol of Stahl and colleagues[37]. In brief, deletion plasmids (see suppl. Table 9) were constructed using the vector pBIISK_*sacB/kanR*, incorporating approximately 1200 bp upstream and downstream flanking regions of the respective target genes. The vector backbone and flanking regions were amplified by PCR employing primers with complementary overhangs and assembled using the NEBuilder HiFi DNA Assembly Kit (NEB, Ipswich, MA, USA) according to the manufacturer's instructions. To prevent potential polar effects on the expression of adjacent genes, the primers (see supplementary Table 10) were designed such that the terminal 30 bp at both the 5′ and 3′ ends of the target genes remained in place after deletion of the internal gene sequence. Transformation of *B. bacilliformis* KC583 with the respective deletion plasmids was performed by electroporation, as described above, with the modification that 40 μg of plasmid DNA (40 μL at 1 μg/μL) was used. Selection of recombinant bacteria with genomic plasmid integration was performed on CBA-kanamycin plates incubated for 10–14 days and confirmed by colony PCR. Subsequent plasmid excision was induced by counter-selection on CBA supplemented with 10% (v/v) sucrose (Thermo Fisher Scientific, Waltham, MA, USA) for an additional 10–14 days. Loss of kanamycin resistance was verified by cultivation on CBA-kanamycin plates, confirming successful plasmid segregation. The deletion of target genes was finally validated by PCR amplification and Sanger sequencing of the respective sequences. The *porA* and *hyd* deletion mutants were complemented with the corresponding genes using the pBBR1MCS-2 vector. To generate the complementation plasmids, the target genes, including approximately 150–300 bp of upstream and downstream regions containing essential transcriptional elements, were PCR-amplified and assembled with the vector. Primers were designed with a BamHI restriction site for subsequent assembly. The amplicons and plasmid DNA were digested with BamHI-HF (NEB), treated with alkaline shrimp phosphatase (NEB) to remove free 5′-phosphate groups, and purified using QIAquick PCR Purification Kits (Qiagen, Hilden, Germany). Ligation (vector:insert ratio 1:5) was performed overnight at 16 °C using T4 DNA ligase (NEB). For double complementation of *porA* and *hyd*, the target genes were PCR-amplified, purified, and assembled into pBBR1MCS2 using the NEBuilder HiFi DNA Assembly Kit (NEB) according to the manufacturer's instructions. Competent *E. coli* NEB 5-alpha cells (NEB) were transformed with the recombinant plasmids and selected on LB agar with kanamycin. Transformants were analyzed by PCR and Sanger sequencing and

correct assembled plasmids were isolated and used to transform the *B. bacilliformis* hemolysis deletion mutants (Δ*porA*, Δ*hyd*) via electroporation. Transformants were selected on CBA-kanamycin plates as described above and verified by Sanger sequencing of the respective gene regions.

## Mutagenesis of the active site of the α/β-hydrolase

Point mutations were introduced into the active site of the α/β-hydrolase using the Q5 Site-Directed Mutagenesis Kit (NEB) according to the manufacturer's instructions. Mutagenic primers were designed using the NEBaseChanger tool and used for PCR amplification (see supplementary Table 10) of the α/β-hydrolase complementation plasmid (pBBR1MCS-2_hyd). The resulting PCR product was treated with the provided kinase, ligase, and DpnI enzyme mix for 5 min at room temperature to phosphorylate and ligate the 5′ ends of the amplicon and to digest the methylated template plasmid. For the triple mutant three fragments with the desired mutation were amplified and assembled using the NEBuilder HiFi DNA Assembly Kit as described above. Subsequently, chemically competent *E. coli* (NEB 5-alpha) cells were transformed with 5 μL of the reaction mixture. Transformants were selected on LB agar plates supplemented with kanamycin and Sanger sequencing was used to confirm the presence of the intended mutation. Verified plasmids carrying the desired point mutations were isolated and used to transform the *B. bacilliformis* α/β-hydrolase deletion mutant (Δ*hyd*). Transformants were selected and isolated as described above.

## Protein expression of a truncated α/β-hydrolase in *B. bacilliformis*

A truncated version of the α/β-hydrolase lacking the hydrophobic C-terminal region was generated by PCR amplification of the hydrolase gene from the pBBR1-MCS plasmid used for *hyd* complementation, excluding the C-terminal hydrophobic domain (amino acids 335–374 in the mature protein). The primers (see supplementary Table 10) contained an overhang compatible with the linearized pBBR1-MCS *hyd* complementation plasmid to facilitate seamless assembly using a Gibson-cloning, as described above. The resulting plasmid was introduced into *E. coli* DH5α by heat-shock transformation, and the resulting plasmid was introduced into a *B. bacilliformis* Δ*hyd* via electroporation as described above. Complementation was confirmed by PCR and Sanger sequencing.

## Quantitative hemolysis assays (hemoglobin, lactate dehydrogenase)

For the quantification of the hemolytic activity of *B. bacilliformis* wildtype and mutants, a hemolysis assay was established by infecting human erythrocytes and quantifying the released hemoglobin via photometric measurement. Human erythrocytes were isolated as described above, washed three times with 0.9% (w/v)NaCl, and the cell number was determined microscopically using trypan blue exclusion. For the hemolysis assay, $10^7$ human erythrocytes were infected with varying MOIs of *B. bacilliformis* (grown for 3 days) and incubated at 37 °C with 5% (v/v) $CO_2$ for 20 h. For negative control, uninfected erythrocytes and for positive control, Triton-X-100 lysed erythrocytes were used. After incubation, the culture medium was separated from the cells by centrifugation at 500 g for 10 min. A 100 μL aliquot of the supernatant was transferred to a flat bottom multi-well plate to determine the free hemoglobin photometrically at 541 nm using a UV spectrophotometer (Sunrise, Tecan, Männedorf, Switzerland).

For determination of free hemoglobin and lactate dehydrogenase (both released from erythrocytes), $10^8$ erythrocytes were infected as described above for 20 h in 48 well plates (Greiner, Bio-One, Frickenhausen, Germany). Supernatants were gently centrifuged for 5 min at 500 g to remove remaining erythrocytes or cell debris and, subsequently, centrifuged through 0.22 μm filters (Merck, Darmstadt,

Germany) at 12,000 g to remove bacteria. The exact amount of free hemoglobin in the supernatants was quantified using the three-wavelength method according to Harboe with a commercially available kit (fHb, Bioanalytic GmbH, Umkirch, Germany) on a Cobas 8000 C502 analyzer (Roche, Mannheim, Germany). Released lactate dehydrogenase (LDH) was measured from the supernatants using the LDHI2 kit (Roche) on a Cobas 8000 C701 IFCC analyzer (Roche), following the manufacturer's instructions.

To determine whether a particular blood group affects susceptibility to or confers protection against *B. bacilliformis*-induced hemolysis, blood samples from donors of different blood groups were infected (see supplementary Fig. 2). The ABO blood groups of the donors were determined by Eldoncard 2511 (Eldon Biologicals A/S, Gentofte, Denmark) following the manufacturer's instructions.

For testing contact dependency of bacterially induced hemolysis, erythrocytes were separated from bacteria using a filter insert (Thincert 0.4 μm, Greiner Bio-One). In particular, $5 \times 10^6$ erythrocytes were resuspended in 500 μL of 0.9% NaCl and placed in the bottom chamber of a 24-well plate (Greiner). The filter insert was positioned above, and $2.5 \times 10^7$ bacteria (resuspended in 100 μL of 0.9% NaCl) were added (MOI: 5). After 20 h of incubation, the filter inserts were carefully removed, and hemolysis was assessed photometrically as described above. Bacterial translocation to the erythrocyte chamber was excluded by plating 100 μL on Columbia agar plates and incubating for two weeks, with no bacterial growth observed. Parallel experiments were performed with bacteria in the well plate and erythrocytes in the filter insert to exclude experimental artifacts. Erythrocyte infections without filter inserts served as controls.

For testing the nature of the hemolytic activity of *B. bacilliformis*, whole cell bacterial lysates were generated by sonication for six cycles each 15 s on ice (Branson Sonifier 450, Danbury, CT, USA). Heat-killed whole cell bacteria were generated by heating at 60 or 95 °C confirmed by the absence of colonies after cultivation on CBA for two weeks.

## RT-PCR analysis
Nucleic acid extraction was performed using RNeasy minikit and DNase-treatment (74104 and 79254, Qiagen, Hilden, Germany) following manufacturer's recommendations. Reverse transcription of extracted RNA was done using LunaScript RT SuperMix kit (E3010, NEB) including a non-RT sample as a control. Amplification of cDNA was performed using specific primers for *porA* and *hyd* and the house keeping gene *rpoD* which was used as a control (see supplementary Table 10).

## Western Blot analysis
Detection of proteins by Western blotting was performed using the Tank-Blot method. Whole cell lysates of $2 \times 10^9$ bacteria (grown for 10 days) were separated by 10% (w/v) SDS-PAGE. The SDS gels were assembled with a nitrocellulose membrane (GE Healthcare, Solingen, Germany) in a blotting apparatus and placed vertically in an electrophoresis chamber filled with transfer buffer (25 mM Tris, 192 mM Glycine (Roth, Karlsruhe, Germany), and 10% (v/v) methanol (Sigma-Aldrich). The proteins were electrophoretically transferred to the nitrocellulose membrane (GE Healthcare) at a constant current of 300 mA for 1 h using a Mini-Trans-Blot cell (Biorad). For immunodetection, the membranes were blocked in blocking buffer (10 mM Tris/HCl pH 7.4, 0.15 M NaCl, 0.2% (v/v) Tween 20 (Roth), 5% (w/v) non-fat milk powder) for 1 h. The membranes were washed with washing buffer (10 mM Tris/HCl pH 7.4, 0.15 M NaCl, 0.2% (v/v) Tween 20) and incubated overnight at 4 °C with rabbit anti porin A or rabbit anti-α/β-hydrolase antibodies at a dilution of 1:500 (see below). After this, the membranes were washed with washing buffer and incubated with the corresponding peroxidase-conjugated polyclonal swine anti-rabbit HRP IgG antibody (Agilent Dako, Santa Clara, CA, USA; number P021702-2) at a dilution of 1:1000 for 1 h at room temperature (RT).

The blots were developed using SuperSignal Chemiluminescence Substrates (Thermo Fisher Scientific) and documented with a ChemiDOC XRS+ System and ImageLab V6.0.1. software (BioRad).

## Generation of peptide-based anti-porin A and α/β-hydrolase-specific antibodies
For generation of antibodies, 15-mer peptide sequences [homologous to specific regions within either porin A (DVDAKTNADLDQHKK) or α/β-hydrolase (VKGKVETNTYKPTHD)] were synthesized and used as antigen (approximately 75 mg/injection) for the generation of rabbit anti-porin A IgG or rabbit anti-α/β-hydrolase IgG antibodies (Eurogentec, Liège, Belgium). Rabbit preimmune sera were used to verify antibody specificity via Western blotting.

## In silico characterization of porin A and α/β-hydrolase
The presumptive pathogenicity factors porin A and α/β-hydrolase were further characterized in silico, following established protocols[51]. To investigate the presence of porin A and α/β-hydrolase in other *B. bacilliformis* strains, a BlastP (National Center of Biotechnology Information[52]) search was performed in the non-redundant protein database. For homology analyses of porin A and α/β-hydrolase, the corresponding amino acid sequences from 14 additional *B. bacilliformis* strains or isolates with closed genomes were extracted and subjected to multiple sequence alignment using Clustal Omega to assess sequence identity[53]. Various physicochemical properties [molecular weight, isoelectric point, extinction coefficient, stability index, aliphatic index, and GRAVY (average hydrophobicity) were analyzed using ProtParam[54]. The subcellular localization of porin A and α/β-hydrolase was predicted using PSORTb Version 3.0.3, CELLO v.2.5, and LocTree3[55–57]. Signal peptide identification was performed using SignalP 6.0[58]. For motif and domain analysis, the software tools NCBI Conserved Domain Search Service (CD Search), Pfam, and InterProScan were employed[59–61]. Protein folding predictions were performed using ColabFold which integrates MMseqs2-based multiple sequence alignment (MSA) with AlphaFold2 structure modeling[62,63]. The corresponding protein sequences (after removal of signal peptides) were used as input. Protein folding predictions were carried out using default settings. The results were visualized using UCSF ChimeraX 1.2.5[64]. Structural homologs of the target α/β-hydrolase were identified using DALI, Cofactor, and PDBeFold[65–67]. The closest structural matches were further analysed in structural alignments using TM-align[23], comparing the AlphaFold-predicted model to the identified homologs.

## Gene cluster comparison of *B. bacilliformis*
Comparative analysis of two gene loci *porA* and *hyd* across 15 closed *B. bacilliformis* genomes was performed using the CAGECAT webserver (Comparative Analysis of Genomic Contexts and Associated Tools), which integrates clinker (v0.0.24) for gene cluster alignment and visualization. The two loci of *por-A* and *hyd* were identified and extracted from annotated genome assemblies[68]. The respective gene clusters were submitted to the CAGECAT interface in GenBank format and aligned with clinker's default parameters. Synteny was assessed across all clusters. Genes were grouped into homology clusters and color-coded accordingly. Identity values between homologous genes were represented by shaded connectors.

## Compound library
To inhibit hemolysis, we systematically examined a library of 27 commercially available phospholipase inhibitors (see supplementary Table 8) composed by Merck (Darmstadt, Germany). Various concentrations of each inhibitor were tested to determine the optimal inhibitory effect. The respective hemolytic activities of *B. bacilliformis* were assessed via the photometric hemolysis assay as described above.

## Data analysis and statistics

Statistical analyses were performed using GraphPad Prism version 6 (GraphPad Software, San Diego, CA, USA). One-way ANOVA was used, followed by Dunnett's post hoc test for comparisons against a control, or Tukey's post hoc test for comparisons among all groups. Data are represented as mean and error bars show standard deviation. A $p < 0.01$ was considered statistically significant.

## Data availability

All data are provided within the manuscript. Transposon mutants, recombinant clones and rabbit antibodies (depending on availability and legal requirements) will be provided upon request by the corresponding author. The data generated in this study are provided in the Source Data file of this paper (Figshare https://doi.org/10.6084/m9.figshare.30165772). Source data are provided with this paper.

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

## Acknowledgements

This work was supported by: (i) LOEWE Center DRUID [(Novel Drug Targets against Poverty-Related and Neglected Tropical Infectious Diseases (project C2)] and the Robert Koch-Institute, Berlin, Germany (*Bartonella* Consiliary Laboratory, 1369-354) both to V.A.J.K, (ii) Suomen syöpäsäätiö (grant number 4709736) and the Academy of Finland (1357076) both to A.G., (iii) the German Research Foundation (grant number AV 9/11-1) to B.A. and (iv) Prociencia Perú (grant PE501096118-2025) to P.T. Funding parties did not influence data analysis, data interpretation, or writing of the manuscript.

## Author contributions

V.A.J.K. conceived the study. A.D., F.W., D.M., W.B., S.B., A.G., and M.G.Q. performed experimental laboratory work, data collection and analysis, data interpretation and writing. A.G., L.S., and P.T. performed bioinformatic analyses, and A.G. and A.D. wrote the section on structural analyses. B.A. and H.B. contributed with protocols and critical review of the manuscript. All authors had full access to all data in the study and had final responsibility for the decision to submit for publication. A.D. and V.K. verified underlying data of the study. All authors approved the final manuscript.

## Funding

## Competing interests

The authors declare no competing interests.
