## [Transparent Peer Review file · Nature Communications]

Porin A and α/β -hydrolase are necessary and sufficient for hemolysis induced by *Bartonella bacilliformis*

Corresponding Author: Professor Volkhard Kempf

Version 0:

Reviewer comments:

Reviewer #1

(Remarks to the Author)

Bartonella bacilliformis is the bacterium that causes Carrión's disease, also known as "Oroya Fever" (acute form) or "Verruga peruana" (Peruvian wart, chronic form), in humans. It is transmitted by the bite of female sandflies of the genus *Lutzomyia*, mainly *L. verrucarum* and *L. peruensis*. This disease is endemic in Peru, Ecuador, and southern Colombia. In Peru, it represents a serious public health problem because it primarily affects rural inhabitants of the inter-Andean valleys and jungle areas, who unfortunately live in poverty and extreme poverty. Although Carrión's disease is currently restricted to those areas, the expansion of the vector's distribution could lead to its spread to other regions in the future. Furthermore, *B. bacilliformis* has been reported to present constitutive resistance to quinolones, one of the drugs used in its treatment. There are reports of persistent cases of Carrión's disease in Peru, underscoring the urgent need to improve control strategies. To achieve this, it is essential to gain more knowledge about the pathogenesis and virulence factors of this bacterium.

In this regard, we believe that the article by Dichter et al. presents results that constitute a very interesting contribution, helping to elucidate the molecular mechanisms by which *B. bacilliformis* induces hemolysis in human erythrocytes, as well as to develop a potential new treatment strategy targeting these virulence factors.

The identification and characterization of porin A and lipase α/β -hydrolase as the virulence factors by which *B. bacilliformis* induces hemolysis is a relevant and original contribution to the knowledge of this pathogenic bacterium.

The methodology applied to generate mutant bartonellae deficient in hemolysis induction by using the Tn5 transposon and complementation with the corresponding genes carried in plasmidial constructs, the mutation directed to the active site of α/β -hydrolase, the hemolysis assays, the evaluation of phospholipase inhibitors and the predicted *in silico* characterization of porin A and α/β -hydrolase, we consider to have been carried out in a solid manner and to have included the necessary controls.

However, we have some major and minor comments that we believe should be clarified by the authors before publication:

Major comments

Line 793: (Figure 1) The data presented for MOI 1 in Panel A do not appear to correlate with the quantitative results shown in Panel B. In Panel A, the visual hemolysis for MOI 1 appears very similar to that of MOI 10, whereas in Panel B, the absorbance values for MOI 1 are substantially lower than those for MOI 10. Please clarify the reason for this discrepancy and explain whether the photographic and spectrophotometric data were obtained from the same samples.

In Figure 1B, a statistically significant difference is reported between the 0.9% NaCl control and MOI 1 ($p < 0.001$). However, the absorbance values at 541 nm appear to be very close to each other. Could the authors clarify how this difference, although statistically significant, is biologically relevant? It would also be important for the authors to provide the exact absorbance values obtained for each condition, so that readers can better assess the magnitude of the effect beyond the *p*-value.

In Figure 4C, several comparisons are reported as highly significant ($p < 0.001$), even though the absorbance values at 541 nm appear very close to each other (e.g., wildtype vs. Δ porA-porA⁺, or Δ hyd vs. Δ hyd-hyd⁺). Could the authors clarify how these differences, although statistically significant, are biologically relevant? Furthermore, it is not clear whether a correction for multiple comparisons was applied, which is essential given the large number of pairwise tests. I also recommend that the authors provide the raw absorbance values (in supplementary material).

Minor comments

Line 144: reads using tryptic soil medium, but it should read using tryptic soy medium.

Line 232: Tecan, Männedorf, Switzerland) For the determination of the AB0 ... (Please clarify the purpose and relevance of this step in the experimental context)

Line 273: *B. bacilliformis* strains, a BlastP (reference)

Line 80: forced endocytosis (reviewed in 9) (Please revise the citation "(reviewed in 9)" for clarity. Consider using a more standard format, such as "as reviewed in [9]" and briefly indicating what aspect is covered in that review)

Reviewer #2

(Remarks to the Author)

Reviewer #3

(Remarks to the Author)

General comments:

The authors convincingly identified PorA and Hyd of *B. bacilliformis* as essential factors in contact-dependent hemolysis. Considering the role of hemolysis in Oroya fever, the acute form of Carrion's disease with mortality rates of up to 90%, this represents a major breakthrough. The indication that Hyd is druggable furthermore provides an avenue for pharmacological intervention against this deadly disease. The manuscript should be considered for publication after revision addressing the points specified below.

Specific comments:

- Figure 2B / pp. 14–15:

The experiment testing for contact-dependent versus diffusible hemolytic activity using a filter with bacteria or RBCs placed on top is not described in sufficient detail—neither in the figure legend nor in the Results section. The best description currently appears in the Discussion (p. 22, lines 517–520). The description in the Materials & Methods section (p. 10, lines 235–240) lacks the necessary detail for reproducibility (e.g., numbers/quantities of bacteria and RBCs, media used, incubation times). This should be clarified and described comprehensively in the Materials & Methods section.

- Figure 4 / p. 16:

It remains unclear why Hyd expression in the Δ porA mutant is undetectable at the protein level (similar to the porA transposon mutant, as noted by the authors), while in the porA/hyd double mutant complemented with the hyd expression plasmid (under its native promoter), Hyd expression is detectable. If this cannot be resolved experimentally, the authors should at least discuss this discrepancy in the Discussion, including its relation to the hemolytic phenotype presented in panel C.

- Figure 8 / p. 20:

The reported dose-dependency of inhibitor 48/80 is unusual, with the strongest inhibitory effect at 10 μ M, while activity decreases at 25 μ M. This raises the possibility that at higher concentrations, the compound itself may exert hemolytic activity (alone, or in combination with *B. bacilliformis* lacking hemolytic activity), which could mask its inhibitory effect. The authors should test this by including control experiments using *B. bacilliformis* mutants deficient in porA and/or hyd.

- Discussion / pp. 24–25:

The authors should expand their discussion to speculate on why hemolysis is contact-dependent in light of current knowledge about the localization and function of PorA and Hyd.

Version 1:

Reviewer comments:

Reviewer #1

(Remarks to the Author)

I have re-reviewed the newly submitted manuscript with the changes made based on the reviewers' comments. I am satisfied with the revisions and believe the manuscript has improved substantially. I believe it is ready for publication and will provide the scientific community with further insights into the pathogenic mechanisms of the bacterium *Bartonella bacilliformis*.

Reviewer #2

(Remarks to the Author)

Reviewer #3

(Remarks to the Author)

The authors have addressed all points raised by this reviewer in a satisfactory manner.

Point-by-point response to referees “Porin A and α/β -hydrolase are necessary and sufficient for hemolysis induced by *Bartonella bacilliformis*”

-revised Nature Communications manuscript

Reviewer #1:

“... we believe that the article by Dichter et al. presents results that constitute a very interesting contribution, helping to elucidate the molecular mechanisms by which *B. bacilliformis* induces hemolysis in human erythrocytes, as well as to develop a potential new treatment strategy targeting these virulence factors. The identification and characterization of porin A and lipase α/β -hydrolase as the virulence factors by which *B. bacilliformis* induces hemolysis is a relevant and original contribution to the knowledge of this pathogenic bacterium. The methodology applied to generate mutant bartonellae deficient in hemolysis induction by using the Tn5 transposon and complementation with the corresponding genes carried in plasmidial constructs, the mutation directed to the active site of α/β -hydrolase, the hemolysis assays, the evaluation of phospholipase inhibitors and the predicted in silico characterization of porin A and α/β -hydrolase, we consider to have been carried out in a solid manner and to have included the necessary controls.”

We thank the reviewer for the positive evaluation.

Major comments

“Line 793: (Figure 1) The data presented for MOI 1 in Panel A do not appear to correlate with the quantitative results shown in Panel B. In Panel A, the visual hemolysis for MOI 1 appears very similar to that of MOI 10, whereas in Panel B, the absorbance values for MOI 1 are substantially lower than those for MOI 10. Please clarify the reason for this discrepancy and explain whether the photographic and spectrophotometric data were obtained from the same samples.

“In Figure 1B, a statistically significant difference is reported between the 0.9% NaCl control and MOI 1 ($p < 0.001$). However, the absorbance values at 541 nm appear to be very close to each other. Could the authors clarify how this difference, although statistically significant, is biologically relevant?”

We think that it is helpful to respond here to both remarks. We obviously described our experiments insufficiently and we would like to explain the experimental setting in more detail.

- (i) The former Fig. 1A and Fig. 1B referred to two independent experimental settings with different MOIs (0.1, 1, 10 and 1, 2, 4, 8, 10) not fitting nicely.
- (ii) In panel A, with increasing MOI an increasing red colour intensity of the supernatant is visible starting at an MOI of 1 whereas in panel B this hemolytic process gets visible at an MOI of 2 (instead of MOI 1).

We have decided to replace this important figure with results from two novel and independent experiments (statement given in Fig. legend 1, line 851) using identical MOIs visible in plastic tubes and quantified by absorbance.

The respective experiments have been expanded as described in Material and Methods and include now hemolysis data as before (**modified Fig.1A**: optical assessment in plastic tubes, **modified Fig. 1B** absorbance values at 541 nm). Additionally, were performed quantitative analyses of free hemoglobin (mg/l) and lactate dehydrogenase (units/l), both released from infected erythrocytes in order to demonstrate the biological relevance and severity of this process (**new suppl. Fig. 1**). Material and Methods have been updated (lines 238-247), the results of quantifying free hemoglobin and lactate dehydrogenase are given in lines 352-355 and the biological relevance is now more broader discussed in lines 524-529. - **Done as suggested.**

"It would also be important for the authors to provide the exact absorbance values obtained for each condition, so that readers can better assess the magnitude of the effect beyond the p-value."

The exact absorbance values for Fig. 1, 2, 4, 7 and 8 are now included in the new **suppl. table 4** and they are directly given in the renumbered **suppl. Figure 2** (formerly **suppl. Figure 1**) and **suppl. Figure 7** (formerly **suppl Figure 6**). These data had also been submitted on the original data file before and they are now given updated in the current original data file. - **Done as suggested.**

"In Figure 4C, several comparisons are reported as highly significant ($p < 0.001$), even though the absorbance values at 541 nm appear very close to each other (e.g., wildtype vs. $\Delta porA-porA^+$, or Δhyd vs. $\Delta hyd-hyd^+$). Could the authors clarify how these differences, although statistically significant, are biologically relevant? Furthermore, it is not clear whether a correction for multiple comparisons was applied, which is essential given the large number of pairwise tests. I also recommend that the authors provide the raw absorbance values (in supplementary material)."

The referee raises an important question regarding the statistical differences observed between the wildtype and the complemented mutants, specifically questioning their biological relevance. We thank the reviewer for this valuable comment, which made us realize that the statistical analyses presented in the previous version of Figure 4, although mathematically correct, are of limited biological utility. In fact, the appropriate comparisons should be as follows: deletion mutants ($\Delta porA$, Δhyd , $\Delta porA/\Delta hyd$) should be compared with the wildtype (with the expected effect: reduced hemolysis), while complemented mutants ($\Delta porA -porA^+$, $\Delta hyd-hyd^+$, $\Delta porA-porA^+/\Delta hyd$, $\Delta porA/\Delta hyd-hyd^+$, $\Delta porA-porA^+/\Delta hyd-hyd^+$) should be compared to their respective deletion mutants (Δpor , Δhyd , $\Delta por/\Delta hyd$; expected effect: restored hemolysis).

Regarding the statistical analysis, corrections for multiple comparisons were consistently applied as described in the Materials and Methods section and are indicated in the figure legends (e.g., Figure legend 4).

The raw absorbance values are now provided in **supplementary Table 4** and referenced in the text. - - **Done as suggested.**

Line 144: reads using tryptic soil medium, but it should read using tryptic soy medium.

We apologize, this mistake has been corrected (line 147). -**Done as suggested.**

"Line 232: Tecan, Männedorf, Switzerland) For the determination of the AB0 ... (Please clarify the purpose and relevance of this step in the experimental context)."

We have clarified the purpose and relevance (line 248-250) as requested. -**Done as suggested.**

"Line 273: B. bacilliformis strains, a BlastP (reference)"

This citation is now given (Citation #27) as requested. -**Done as suggested.**

"Line 80: "forced endocytosis (reviewed in 9) Please revise the citation "(reviewed in 9)" for clarity. Consider using a more standard format, such as "as reviewed in [9]" and briefly indicating what aspect is covered in that review."

We have included the requested information (lines 80-83) and revised the citation phrase (line 83)."
-**Done as suggested.**

Reviewer #3 (Remarks to the Author):

"The authors convincingly identified PorA and Hyd of *B. bacilliformis* as essential factors in contact-dependent hemolysis. Considering the role of hemolysis in Oroya fever, the acute form of Carrion's disease with mortality rates of up to 90%, this represents a major breakthrough. The indication that Hyd is druggable furthermore provides an avenue for pharmacological intervention against this deadly disease. The manuscript should be considered for publication after revision addressing the points specified below."

We thank the reviewer for the positive evaluation.

"Figure 2B / pp. 14-15: The experiment testing for contact-dependent versus diffusible hemolytic activity using a filter with bacteria or RBCs placed on top is not described in sufficient detail—neither in the figure legend nor in the Results section. The best description currently appears in the Discussion (p. 22, lines 517-520). The description in the Materials & Methods section (p. 10, lines 235-240) lacks the necessary detail for reproducibility (e.g., numbers/quantities of bacteria and RBCs, media used, incubation times). This should be clarified and described comprehensively in the Materials & Methods section."

We apologize, this information is now given in detail (lines 253-261). - ***Done as suggested.***

"Figure 4 / p. 16: It remains unclear why Hyd expression in the Δ porA mutant is undetectable at the protein level (similar to the porA transposon mutant, as noted by the authors), while in the porA/hyd double mutant complemented with the hyd expression plasmid (under its native promoter), Hyd expression is detectable. If this cannot be resolved experimentally, the authors should at least discuss this discrepancy in the Discussion, including its relation to the hemolytic phenotype presented in panel C."

This discrepancy is difficult to explain and, as we do not have any experimental clue to solve this phenomenon, we follow the recommendation of the referee and speculate on this phenomenon (lines 572-577). - ***Done as suggested.***

"Figure 8 / p. 20: The reported dose-dependency of inhibitor 48/80 is unusual, with the strongest inhibitory effect at 10 μ M, while activity decreases at 25 μ M. This raises the possibility that at higher concentrations, the compound itself may exert hemolytic activity (alone, or in combination with *B. bacilliformis* lacking hemolytic activity), which could mask its inhibitory effect. The authors should test this by including control experiments using *B. bacilliformis* mutants deficient in porA and/or hyd."

We thank the referee for this important remark and repeated the experiments accordingly adding *B. bacilliformis* Δ hyd. In fact, the deletion mutant did not show hemolytic activity and compound 48/80 increased erythrocyte hemolysis only slightly at 25 μ M excluding unspecific hemolytic effects of compound 48/80. This information is described in the modified Figure 8 and within the text in lines 506-508.- ***Done as suggested.***

"Discussion / pp. 24-25:" The authors should expand their discussion to speculate on why hemolysis is contact-dependent in light of current knowledge about the localization and function of PorA and Hyd."

We have included a respective paragraph in the discussion (lines 612-621). - ***Done as suggested.***